# Casein-Based Nanoparticles: A Potential Tool for the Delivery of Daunorubicin in Acute Lymphocytic Leukemia

**DOI:** 10.3390/pharmaceutics15020471

**Published:** 2023-01-31

**Authors:** Nikolay Zahariev, Milena Draganova, Plamen Zagorchev, Bissera Pilicheva

**Affiliations:** 1Department of Pharmaceutical Sciences, Faculty of Pharmacy, Medical University of Plovdiv, 15A Vassil Aprilov Blvd, 4002 Plovdiv, Bulgaria; 2Research Institute, Medical University of Plovdiv, 15A Vassil Aprilov Blvd, 4002 Plovdiv, Bulgaria; 3Department of Medical Biology, Medical Faculty, Medical University of Plovdiv, 15A Vassil Aprilov Blvd, 4002 Plovdiv, Bulgaria; 4Department of Medical Physics and Biophysics, Faculty of Pharmacy, Medical University of Plovdiv, 15A Vassil Aprilov Blvd, 4002 Plovdiv, Bulgaria

**Keywords:** casein, nanoparticles, daunorubicin, acute lymphocytic leukemia, targeted drug delivery, nano spray-drying, Reh cell line, cytotoxicity

## Abstract

The aim of this study was to develop casein-based nanoscale carriers as a potential delivery system for daunorubicin, as a pH-responsive targeting tool for acute lymphocytic leukemia. A coacervation technique followed by nano spray-drying was used for the preparation of drug-loaded casein nanoparticles. Four batches of drug-loaded formulations were developed at varied drug–polymer ratios using a simple coacervation technique followed by spray-drying. They were further characterized using scanning electron microscopy, dynamic light scattering, FTIR spectroscopy, XRD diffractometry, and differential scanning calorimetry. Drug release was investigated in different media (pH 5 and 7.4). The cytotoxicity of the daunorubicin-loaded nanoparticles was compared to that of the pure drug. The influence of the polymer-to-drug ratio on the nanoparticles’ properties such as their particle size, surface morphology, production yield, drug loading, entrapment efficiency, and drug release behavior was studied. Furthermore, the cytotoxicity of the drug-loaded nanoparticles was investigated confirming their potential as carriers for daunorubicin delivery.

## 1. Introduction

One of the leading causes of death worldwide is cancer and its related complications. By the end of the 20th century, the most common therapeutic approach to cancer treatment was surgery [1]. Nowadays, along with surgery, chemotherapy is considered the basis for the treatment of the most common cancers (breast, lung, liver, prostate, stomach, cervical, and thyroid cancers), including hematological malignancies [2]. Despite their high therapeutic efficacy, conventional chemotherapeutics are characterized by a lack of selectivity for accumulation in the tumor cells and by a number of serious side effects observed during treatment. These side effects are related to inadequate biodistribution of the chemotherapeutics, which leads to accumulation in healthy tissues. Therefore, one of the main challenges facing chemotherapy is the development of drug delivery systems that are effective and provide high selectivity, minimizing side effects and achieving maximum antitumor effect [1]. In recent decades, the advancement in the field of nanotechnology and oncology has led to the development of innovative chemotherapeutic drug delivery systems for targeted therapy. Such systems can specifically target tumor cells without affecting healthy tissues [3]. However, to be used as targeting drug carriers, these systems must meet certain criteria regarding their structure, particle size, biocompatibility, toxicity, and immunogenicity. Effective chemotherapeutic delivery requires the carrier to remain stable in the circulation, then pass across specific biological membranes and reach the target site. Moreover, it should be able to avoid clearance by the reticuloendothelial system and achieve a high degree of drug accumulation in the tumor tissue or specific binding to target cells, which can be accomplished either by active or passive targeting [4].

Advances in nanotechnology have led to the possibility of developing drug delivery systems based on polymer nanoparticles, which have the property to target specific tissues and therefore reduce dose-dependent side effects. Over the last decades, biopolymers have been the subject of serious scientific interest due to their specific characteristics and numerous advantages over synthetic polymers such as low immunogenicity, high biocompatibility, biodegradability, and low toxicity [5]. Biopolymers such as peptides and polysaccharides have been widely used to formulate nanoparticulate drug delivery systems that can provide controlled and targeted release, improving the biodistribution of the encapsulated drug and minimizing its side effects [6]. Compared to polysaccharides, proteins are preferred as structure-forming biopolymers for targeted delivery because of their ability to produce small-sized particulate systems with enhanced cell penetration [5,7]. Proteins can bind to different active molecules, protect them and control their release at the desired site within the body. There are numerous reports for the use of proteins such as gelatin, collagen, whey proteins, and casein for the preparation of microparticles, nanoparticles, or hydrogel delivery systems for drugs, nutrients, bioactive peptides, and probiotic organisms. Due to their nutritional and functional properties, milk proteins are widely used in the food and pharmaceutical industry [8,9].

Among the proteins used for the formulation of nanoparticulate carriers, casein has been found to have optimal structural and physicochemical properties for the development of targeted drug delivery systems with pH-dependent behavior. Casein, one of the major milk proteins, is biodegradable, non-toxic, highly stable, and inexpensive. Surfactant and stabilizing properties, the ability to bind ions and small molecules, as well as emulsification, self-assembly, and pH-responsive gelation properties make casein a suitable biopolymer for the development of targeted drug delivery systems [8]. Casein from bovine milk consists of four amphiphilic peptides—αs1, αs2, β, and k, which have a molecular weight between 19 and 25 kDa and an isoelectric point (pI) between 4,6 and 4,8 [10]. Due to their amphiphilic properties, the tendency to bind calcium phosphate and the formation of electrostatic, hydrogen, and hydrophobic interactions, caseins can self-assemble and form casein micelles with an average diameter of 50 to 500 nm [11]. Micelles exhibit pH-dependent behavior because their structure tightens with the decreasing negative surface charge of casein molecules and expands with increasing surface charge, leading to electrostatic repulsion between molecules [12,13,14].

Given the high degree of biocompatibility and biodegradability, the low extent of immunogenicity and toxicity of casein, as well as its pH-dependent behavior and amphiphilic properties, it is obvious why this natural polymer is preferred for the preparation of chemotherapeutic nanoparticulate drug delivery systems.

Various methods have been reported for the preparation of chemotherapeutic-loaded casein nanoparticles, including the self-assembled micellization/lyophilization technique [15], emulsification–ionic gelation [16,17], emulsion-precipitation [18], self-assembled micellization/dialysis [19,20,21], self-micellization/spray-drying [16,22,23], polyelectrolyte complexation [24], non-covalent polymerization [25], and Maillard conjugation [26].

A commonly used method for the preparation of protein nanoparticles for drug delivery is spray-drying. However, the separation and collection of such particles is known to be extremely challenging due to the low collection efficiency for fine particles smaller than 2 μm. To improve the effectiveness of the spray-drying process and to generate mainly nanosized particles, Swiss Büchi Labortechnik AG developed a revolutionary spray-dryer (Nano Spray-dryer B-90) [27]. Thanks to the innovative piezoelectric vibrating spray mesh head, which allows for the formation of fine droplets, and the novel electrostatic particle collector, it is possible to generate particles in the size range from 300 nm to 5 μm. Spray-drying of protein nanoparticles has been reported in numerous scientific papers [28,29,30,31], but no data were found in the literature on the nano spray-drying of casein nanoparticles loaded with chemotherapeutic agents. In this study, the encapsulation of the chemotherapeutic agent daunorubicin hydrochloride (DRB) in casein nanoparticles was demonstrated using a nanospray-drying technique.

Anthracycline antibiotics are among the most effective chemotherapeutic agents and are still considered major components of first-line chemotherapy in the treatment of solid and hematological malignancies [32]. Daunorubicin, an anthracycline antibiotic that possesses antineoplastic activity, is commonly used to treat acute myeloid leukemia (AML) and acute lymphocytic leukemia (ALL) [33]. Its antimitotic and cytotoxic effects result from the generation of free radicals and the inhibition of cell reproduction by the formation of complexes with DNA. The interaction between the base pairs inhibits the activity of topoisomerase II by stabilizing the DNA topoisomerase II complex, preventing the ligation–religation reaction that topoisomerase II catalyzes [34]. Despite its high activity, clinical use of DRB is limited due to multiple drug resistance (MDR) and its serious side effects, including cardiotoxicity, myelosuppression, and the occurrence of oral ulcers [33]. MDR results from the overexpression of P-glycoprotein efflux pumps on the surface of cancer cell membranes. This protein makes cells resistant to drug therapy by reducing the amount of drug entering cancer cells. Technological strategies for tumor-targeted delivery and thus reduced side effects can be accomplished in two ways: by the development of nanosized carriers that improve biodistribution within tumor tissues or by conjugation of anthracyclines to a carrier that specifically binds to ligands overexpressed on the cell membrane of the tumor cells [32]. 

The aim of this study was to develop nanosized casein-based carriers of daunorubicin, as a potential delivery tool for acute lymphocytic leukemia.

## 2. Materials and Methods

Daunorubicin hydrochloride (Mw 563.98 g/mol), sodium caseinate (from bovine milk), CaCl_2_·2H_2_O (Mw 147.01 g/mol), phosphoric acid 85%, sodium dodecyl sulfate, penicillin, streptomycin, fetal bovine serum (FBS), MTT (3-(4,5-dimethylthiazol-2-yl)-2,5-diphenyltetrazolium bromide), poly-D-lysine, and dimethyl sulfoxide (DMSO) were purchased from Sigma-Aldrich (St. Louis, MO, USA). The acetonitrile (HPLC grade) was supplied from Fisher Chemical, Lough-borough, UK. The Reh cell line (ATCC CRL-8286) was kindly provided by Dr. Tino Schenk (University of Jena). The RPMI-1640 culture medium was purchased from PAN Biotech, Aidenbach, Germany. Seahorse XFp Base medium (pH 7.4) was supplied by Agilent (Santa Clara, CA, USA). All other reagents were of analytical grade. 

### 2.1. Preparation of DRB-Loaded Casein Nanoparticles

The coacervation technique followed by spray-drying using a Büchi B-90 nano spray-dryer (Büchi Labortechnik AG, Flawil, Switzerland) was employed for the preparation of drug-loaded casein nanoparticles. Casein solutions were prepared at varied concentrations by dissolving sodium caseinate in deionized water (pH 5.8). After the complete dissolution of sodium caseinate, 5 mg daunorubicin hydrochloride was added. The mixture was stirred on a magnetic stirrer for 60 min in order to achieve maximum incorporation of the drug. Then, the crosslinking agent calcium chloride dihydrate (2 µL/mL) was added dropwise under high-speed homogenization at 25,000 rpm (Miccra MiniBatch D-9, MICCRA GmbH, Heitersheim, Germany) for 15 min, which resulted in the formation of casein micelles. Finally, the obtained nanosuspension of casein micelles was spray dried using a Büchi B-90 nano spray-dryer under the following conditions: a mesh size of 4.0 μm, an inlet temperature of 40 °C, a solution feed rate of 60%, a spray intensity of 60%, a drying gas speed of 120 L/min, and a pressure of 30 nbar. Four batches of drug-loaded formulations were developed at varied drug–polymer ratios. The composition of the batches is presented in Table 1.

### 2.2. Characterization of the Obtained DRB-Loaded Nanoparticles

#### 2.2.1. Production Yields, Drug Loading, and Entrapment Efficiency

The production yields of the obtained nanoparticles from different batches were calculated according to Equation (1):(1)$$\mathrm{Production}\;\mathrm{yield}\;\left(\%\right)=\frac{\mathrm{Spray}\;\mathrm{dried}\;\mathrm{nanoparticles}\;\left(\mathrm{mg}\right)}{\mathrm{Drug}\;\left(\mathrm{mg}\right)+\mathrm{Polymer}\;\left(\mathrm{mg}\right)}\times 100$$

Drug loading (DL) and entrapment efficiency (EE) were determined by HPLC analysis. Ten milligrams of each batch of nanoparticles were dissolved in 5 mL mobile phase and sonicated for 5 min into an ultrasonic bath (Sonorex Bandelin electronic, Berlin, Germany) until dissolution of the particles and extraction of DRB was achieved. The blend was then centrifuged at 5000 rpm for 15 min and filtered (0.22 µm, Chromafil^®^, Macherey-Nagel, Düren, Germany). The supernatant was recovered and the concentration of DRB was determined via HPLC (UltiMate 3000, Thermo Scientific, Waltham, MA, USA) under the following conditions: a 254 nm wavelength; a 5 µL injection volume; an Inertsil^®^ ODS-3 HPLC column of 5 µm, 150 × 4.6 mm (GL Sciences, Tokyo, Japan); a 1 mL/min flow rate; a 25 °C oven temperature; a 20 °C sampler temperature; and mobile phase: 0.29% sodium dodecyl sulfate in 0.2% aqueous solution of phosphoric acid:acetonitril (1:1). The drug loading and entrapment efficiency were calculated using Equations (2) and (3), respectively.
(2)$$\mathrm{DL}\;\left(\%\right)=\frac{\mathrm{Drug}\;\mathrm{amount}\;\mathrm{in}\;\mathrm{the}\;\mathrm{formulation}}{\mathrm{Amount}\;\mathrm{of}\;\mathrm{nanoparticles}}\times 100$$
(3)$$\mathrm{EE}\;\left(\%\right)=\frac{\mathrm{Actual}\;\mathrm{drug}\;\mathrm{content}}{\mathrm{Theoretical}\;\mathrm{drug}\;\mathrm{content}}\times 100$$

#### 2.2.2. Particle Size Analysis, Size Distribution, and Zeta Potential

The particle size of the DRB-loaded casein nanoparticles was analyzed by dynamic light scattering (Microtrac, York, PA, USA). The system measures the particle size in the range of 0.8 nm to 6.5 µm with non-invasive backscattering technology using a 3 mW helium/neon laser at a 780 nm wavelength. The system allows measurements of ζ-potential in the range from −200 mV to +200 mV. All analyses were performed in triplicates at 25 °C.

#### 2.2.3. Scanning Electron Microscopy

Visualization of the obtained particles was performed using scanning electron microscopy (Prisma E SEM, Thermo Scientific, Waltham, MA, USA). The samples were loaded on a copper sample holder and sputter coated with carbon followed by gold using a vacuum evaporator (BH30). The images were recorded at 15 kV acceleration voltage at various magnifications using an ETD (Everhart–Thornley) detector. 

#### 2.2.4. Thermogravimetry Differential Thermal Analysis (TG/DTA)

For the thermal analysis, 3–6 mg samples were placed in Al_2_O_3_ crucibles and heated from 20 to 250 °C, in a static air atmosphere, with a heating rate of 10 °C min^−1^ using a SetSYS-2400 calorimeter system (Setaram Instrumentation, Caluire-et-Cuire, France).

#### 2.2.5. X-ray Diffraction (XRD)

The degree of crystallinity of the synthesized nanoparticles was studied by X-ray powder diffractometry. The diffraction patterns of blank and DRB-loaded casein nanoparticles were recorded at a 2*θ* range from 5° to 90° using an Empyrean X-ray powder diffractometer (Malvern Panalytical, Malvern, UK), equipped with a TTK 600 low-temperature chamber (Anton Paar, Graz, Austria). All the measurements were performed at a voltage of 40 kV and 30 mA.

#### 2.2.6. Fourier Transform Infrared Spectroscopy (FTIR)

FTIR spectroscopy was used to investigate possible drug/polymer interactions. The spectra were collected in the range from 600 cm^−1^ to 4000 cm^−1^ with a resolution 4 nm and 16 scans, using a Nicolet iS 10 FTIR spectrometer (Thermo Fisher Scientific, Pittsburg, PA, USA). The instrument is equipped with a diamond attenuated total reflection (ATR) accessory and the spectra were analyzed with the OMNIC^®^ software package (Version 7.3, Thermo Electron Corporation, Madison, WI, USA).

### 2.3. In Vitro Drug Release

In vitro release of DRB from the casein nanoparticles was performed using the dialysis bag method. A dialysis membrane (Sigma, MWCO 12 kDa) was hydrated in distilled water for 24 h. An accurately weighed amount of nanoparticles (equivalent to 1.5 mg DRB) was dispersed in 5 mL from dissolution media and then transferred to the dialysis bag, which was closed using a plastic clamp. Each bag was placed into a beaker containing 20 mL of each fresh dissolution medium and kept under constant mild agitation on an electromagnetic stirrer at (37.0 ± 0.5) °C. Aliquots (2 mL) were taken and subsequently replaced at predetermined time intervals with fresh medium. Each sample was filtered through a 0.45 μm membrane filter (Chromafil^®^, Macherey-Nagel, Düren, Germany) and the pH was adjusted to pH 2 using phosphoric acid. The analysis for drug content was performed by HPLC as described in Section 2.2.1. The drug release study was performed in two different dissolution media (PBS pH 5 and PBS pH 7.4) for 30 h. The mean results of triplicate measurements and standard deviations were reported.

### 2.4. In Vitro Cytotoxicity

#### 2.4.1. Cell Culture

Lymphoblastic cell line Reh (CRL-8286™, ATCC, Manassas, VA, USA), isolated from a patient with acute lymphocytic leukemia, was cultivated in RPMI-1640 media containing 10% FBS, 1% penicillin/streptomycin in cell culture incubator at (37.0 ± 0.5) °C, 5% CO_2_, and high humidity. The plates were pre-coated with poly-D-lysine to ensure cell adhesion. The cells were seeded at a density of 2 × 10^5^ cells per well into 96-well plates and in Seahorse XF plates depending on the required analysis. The cells were then treated with DRB-free and DRB-loaded casein nanoparticles and incubated for 3, 24, or 72 h, after which the assays were performed. Prior to the analyses, the cells were observed under an inverted microscope to ensure that a monolayer was formed. 

#### 2.4.2. MTT Assay

After different incubation periods, an MTT cytotoxicity test was performed. The MTT dye (3-(4,5-dimethyl-2-thiazolyl)-2,5-diphenyl-2H-tetrazolium bromide) was dissolved in PBS and added to each of the 96 wells to obtain a final concentration of 500 µg/mL After a 2 h incubation process at (37.0 ± 0.5) °C, 100 µL of the extracting agent DMSO was added. The results were measured spectrophotometrically at a 620 nm wavelength on an ELISA reader and presented as a percentage (%) of the vitality of treated cells compared to control cells. Each experiment was triplicated, and the results were calculated as mean values.

#### 2.4.3. Analysis of Mitochondrial Function

To assess mitochondrial function, a Mito stress test on a Seahorse XFp analyzer (Agilent, Santa Clara, CA, USA) was performed. The entire culture media was replaced with Seahorse XFp Base medium (pH 7.4). Following the manufacturer’s protocol, the mitochondrial inhibitors oligomycin, FCCP (trifluoromethoxy carbonyl cyanide phenylhydrazone), and rotenone/antimycin A were consecutively injected into the ports of the cartridge. Injection of the first chemical, known as an ATP synthase inhibitor, determines the production of ATP. FCCP makes the inner mitochondrial membrane permeable to protons and allows maximum electron flux through the electron transport chain (ETC). Thus, collapse of the inner membrane gradient is induced, driving mitochondria to respire at the maximum rate. The latter of the chemicals directly inhibits Complex I of the ETC and shuts mitochondrial respiration down (Figure 1). Such a combination of inhibitors allows for the assessment of several parameters: basal and maximal respiration, ATP levels, and spare respiratory capacity, as well as other mitochondrial characteristics such as proton leak, and non-mitochondrial respiration (Figure 2). For each experiment, the oxygen consumption rate (OCR) was measured, and the data were analyzed using Seahorse Wave Desktop Software v2.6.0 (Agilent, Santa Clara, CA, USA). 

### 2.5. In Vitro Fluorescence Microscopy

To investigate the cellular penetration and intracellular distribution of DRB, fluorescence imaging was performed with a fluorescence microscope Axio Scope A1 (Carl Zeiss Microscopy GmbH, Jena, Germany). Prior to microscopic analysis, the cells were fixed as follows: after treatment for 3 h and 24 h, the Reh cells were washed twice with PBS, centrifuged, and the cell pellet was treated with 4% paraformaldehyde. After an additional centrifugation step, it was redispersed in PBS and the cells were placed on microscopic slides, covered with coverslips, and excited at 545 nm using a 50 W Hg lamp. The samples were observed at ×400 magnification using a filter of 570 nm.

### 2.6. Statistical Analysis

Pre-processing of raw OCR and ECAR data was carried out using Seahorse XF Mito Stress report generators [36]. GraphPad Prism v.6 (GraphPad Software, San Diego, CA, USA) was used for figure preparation and statistical analysis for Seahorse and MTT data. One-way ANOVA followed by a Bonferroni post-hoc test and a two-tailed t-test for multiple group comparisons were used. Probability levels of 0.05 were considered as statistically significant (*p* < 0.05). The statistical analysis represents the data as a mean value ± SEM (SEM—standard error of the mean).

## 3. Results and Discussion

### 3.1. Synthesis and Characterisation of DRB-Loaded Casein Nanoparticles

Casein nanoparticles loaded with DRB were synthesized using a simple coacervation technique followed by spray-drying. The influence of the polymer-to-drug ratio on the nanoparticles’ properties such as their particle size, surface morphology, production yield, drug loading, entrapment efficiency, and drug release behavior was studied. For this purpose, four batches of drug-loaded nanoparticles were prepared and further characterized. The results are presented in Table 2. The nanoparticle production yields ranged widely from 37.67% for batch Cas4-DRB-5 to 81.12% in batch Cas1-DRB-5, steadily increasing when lower casein concentrations were used. Probably, it was the larger amount of polymer in Cas4-DRB-5 that led to increased viscosity of the feeding suspension, hindering free passage through the spray mesh of the spray-dryer. Another plausible explanation for this phenomenon could be the disruption of micellar integrity due to the displacement of calcium phosphate and the formation of precipitate prior to spray-drying [37].

The drug entrapment efficiency ranged from 42.80% to 61.80% and the drug loading was in the range from 2.14% to 3.09%. Considering the polymer-to-drug ratios during formulation, such low values for drug loading were not unexpected. The lowest values for drug loading and entrapment efficiency were established in batch Cas4-DRB-5, which was prepared at a 100:1 ratio. It could be assumed that due to the high concentration of casein (2%), the concentration of calcium chloride as a crosslinker was not sufficient for effective crosslinking, leading to the formation of unstable structures. Morphological analysis performed by scanning electron microscopy confirmed this hypothesis (Figure 3). No clearly defined structures were observed in the scanning electron micrographs. On the contrary, numerous agglomerates with irregular structures were found. A similar phenomenon was reported by Sinaga and co-workers and was thoroughly described [38]. Decreasing the polymer-to-drug ratio from 100:1 to 25:1 resulted in an almost two-fold increase in DL and EE (Table 2). Reduction of the casein concentration from 2% to 0.5% at a constant calcium chloride concentration allowed more efficient polymer crosslinking, leading to the formation of well-defined stable structures with DL of 3.09% and EE of 61.80% (batch Cas1-DRB-5; 25:1 ratio). The latter suggests that not only the polymer concentration and the polymer-to-drug ratio but also the polymer/crosslinker ratio determine the main characteristics of the formulated nanoparticles.

The median size of the obtained particles ranged from 127 nm to 167 nm (Table 2) indicating a clear tendency for size reduction with the decrease in polymer concentration. The histogram of batch Cas4-DRB-5 (Figure 4) displays a bimodal particle size distribution suggesting the presence of aggregates, as hypothesized from DL and EE determination. The zeta potential of the four batches ranged from −18.63 mV to −33.21 mV, indicating the formation of a stable colloid suspension (Table 2). The negative charges on the surface of the particles resulted from the net electrostatic charge at pH values above the isoelectric point (4.6), where the casein carboxylic groups become negatively charged [39].

The surface morphology of the obtained particles was studied using scanning electron microscopy (Figure 3). Two types of morphologies were observed, namely solid dense and toroidal (doughnut-shaped) particles. Solid dense particles were formed at higher concentrations of the polymer—2% and 1.5% (batches Cas4-DRB-5 and Cas3-DRB-5, respectively). Generally, higher concentrations of polymer solution lead to the formation of denser droplets, due to the increased surface energy inside the droplets, which in turn results in the formation of more stable particles [36]. The results were supported by the DLS analysis where these two batches demonstrated zeta potential values −33.21 mV and −25.53 mV, respectively. In batches Cas2-DRB-5 and Cas1-DRB-5, probably due to the formation of smaller droplets, the airflow rate led to destabilization of the droplet shape and the formation of toroidal particles upon solvent evaporation. This type of morphology may also be attributed to the larger amount of water that must be evaporated in the spray-drying process. Evaporation of water from the surface of the generated droplets into the gas stream can cause thermophoretic displacement of the particles inside the droplet, leading to stronger deformation of the droplet and the formation of toroid-shaped particles [40]. 

The physical state of the drug before and after incorporation in the casein nanoparticles was investigated using TG/DTA. Casein thermogram displayed a broad endothermic peak at around 85 °C due to the evaporation of water from the sample (Figure 5). Daunorubicin, on the other hand, being crystalline in nature, is characterized by its melting point at around 180 °C. The DRB-free casein nanoparticles exhibited a sharp endothermic peak at 178.79 °C. After incorporation of the drug into the nanoparticles, the sharp peak corresponding to the melting point of DRB disappeared and the thermograms resembled those of the blank casein nanoparticles, which implies molecular dispersing of the drug into the protein matrix. Similar results were reported by Elzoghby et al. [16] and Puthli et al. [41]. 

The results were confirmed by the XRD study (Figure 6). The XRD pattern of pure DRB showed several characteristic peaks in the 2*θ* region—3.18°, 7.89°, 11.42°, 21.28°, and 24.42°, resulting from its crystalline structure. These peaks were not observed in any of the batches of DRB-loaded casein nanoparticles. Moreover, the typical amorphous XRD pattern of casein was observed, indicating transformation of the drug from a crystalline to an amorphous state. Similar results have been reported after spectroscopic studies on the interaction of the γ-cyclodextrin–daunorubicin inclusion complex [42].

In the FTIR spectra of blank casein nanoparticles (Figure 7), two distinct peaks were found at 1646 cm^−1^ in the amide I region and 1530 cm^−1^ in the amide II region, corresponding to the stretching of the carbonyl group (C=O) and to the symmetric stretching of N-C=O bonds, respectively. The peak at 1077 cm^−1^ could be due to interactions of monocationic phosphates with Na^+^ while the band at 977 cm^−1^ corresponds to bionic phosphate, confirming interactions with Ca^2+^. Two characteristic peaks of DRB were observed in the spectrum of drug-loaded casein nanoparticles. The first peak is shown at 1616.48 cm^−1^ due to C=O stretching of hydrogen-bonded quinone carbonyl groups, and the second peak is at 1581.44 cm^−1^ and attributed to C=C stretching [43], indicating successful drug incorporation into the polymer matrix. In all of the batches of drug-loaded nanoparticles, no shifts in the characteristic peaks of DRB and casein were observed, indicating the absence of interactions between the molecules.

### 3.2. In Vitro Drug Release

To simulate the physiological conditions, the drug release study was carried out in a neutral medium (PBS, pH 7.4). Additionally, the test was performed in a slightly acidic medium (PBS, pH 5) to mimic the environment of tumor cells and to investigate how pH might affect the release rate of DRB from the nanoparticles.

All four batches showed prolonged release profiles in both PBS buffers (Figure 8). At pH 7.4, incomplete drug release (43.73% ± 4.72%) was observed during the first 6 h from the batch Cas4-DRB-5, which was probably due to the higher polymer content and hindered drug diffusion. A similar release pattern was found for batch Cas3-DRB-5 (38.57% ± 7.45%). In batch Cas2-DRB-5, the trend was maintained (26.53% ± 3.19% within 6 h), but additional amounts of DRB were released between hours 9 and 24, which was not observed in the previous batches. The batch Cas1-DRB-5 showed higher release after 6 h (44.02% ± 3.05%) compared to the other three batches, which may be related to higher drug loading and the accumulation of DRB in the periphery of the nanoparticles in the process of spray-drying. Furthermore, DRB release was biphasic with a clearly distinguished burst release phase (20.17% ± 3.80% within the first hour) followed by sustained release of more than 70% after 24 h of testing (71.10% ± 2.22%). A statistically significant difference (*p* < 0.01) was observed in the released drug between 6h and 24 h for batches Cas2-DRB-5 and Cas1-DRB-5, whereas for the batches Cas3-DRB-5 and Cas4-DRB-5, the difference was insignificant, suggesting that the polymer-to-drug ratio played a key role in drug incorporation mode. Probably, at high casein-DRB ratios (100:1 and 75:1), the drug was incorporated deep into the core of the nanoparticles unable to undergo diffusion through the polymer and ensure complete drug release. On the other hand, when larger amounts of DRB were used (50:1 and 25:1), the drug molecules were probably distributed evenly in the internal core and the periphery of the nanoparticles allowing a more complete dissolution for the same test period. Moreover, the results could be associated with the smaller size of these particles and the shorter path for drug diffusion within the casein matrix. Cumulative drug release at pH 7.4 was analyzed for statistically significant differences between the four batches. No statistically significant difference was observed in the released DRB between the samples at either hour 1 or hour 3. A slight decrease was demonstrated for batch Cas2-DRB-5 at hour 6 (26.53% ± 3.19%) compared to the other three batches which released about 40% on average. The trend persisted at hour 9, after which drug release from batches Cas3-DRB-5 and Cas4-DRB-5 stopped, the cumulative DRB was about 55% on average, and Cas1-DRB-5 was the only sample that continued to release the drug and showed a statistically significant difference in DRB release at 24 h. Based on these results, batch Cas1-DRB-5 was considered optimal in terms of in vitro biopharmaceutical performance, whereas Cas2-DRB-5 did not meet the desired release pattern.

At pH 5, similar drug release patterns were obtained. DRB release was incomplete and delayed by about 3 h compared to DRB release at pH 7.4. For example, batch Cas4-DRB-5 released 43.73% ± 4.72% after 6 h at pH 7.4 whereas a similar amount (48.86% ± 6.00%) was released from the same batch at pH 5 after 9 h. The same trend was observed with batch Cas3-DRB-5 (38.57% ± 7.45% at pH 7.4/6 h vs. 38.45% ± 4.84% at pH 5/9 h). Batch Cas2-DRB-5 followed similar release patterns with only 33.79% ± 2.70% DRB released after 9 h of study. All three samples showed no significant change in DRB release after 9 h. The only model that continued to release DRB in the 9–24 h interval was Cas1-DRB-5. Moreover, the release profile was smooth and lacked an initial burst release. DRB release was incomplete, reaching 39.50% ± 12.10% after 9 h, but it was the only sample that demonstrated statistically significant drug release in the 9–24 h interval, with it reaching 54.90% ± 9.95% DRB release. Comparison of the cumulative drug release at pH 5 between the four batches at the same time points revealed a statistically significant difference only for sample Cas4-DRB-5. This batch demonstrated the highest drug release rate (almost 30% DRB was released within 1 h and the trend was maintained until the end of the study).

Statistical analysis of drug release patterns obtained at different pH values (7.4 and 5) showed no significant difference in the DRB release behavior for batches Cas4-DRB-5, Cas3-DRB-5, and Cas2-DRB-5 regardless of the sampling time for both of the dissolution media tested (7.4 or 5). The only standout batch was Cas1-DRB-5, which demonstrated differences in drug release behavior immediately after the first hour of the study. This result confirmed the potential of this batch for further research and application for drug delivery.

It can be assumed that the difference in the release profiles at the pH values studied is due to alterations in the structure and size of the particles. It is well known that casein particles have a pH-dependent swelling behavior [44,45]. At a pH close to the isoelectric point of casein (4.6), hydrophobic interactions lead to a reduction in repulsion between casein molecules, causing the formation of more compact structures with reduced size, leading to more controlled release of DRB [46,47]. Increasing the pH to 5 does not result in a swelling effect [48], but further alkalization leads to a gradual change in particle size due to loss of structural integrity. This effect was most evident for small particles (batch Cas1-DRB-5), suggesting a relationship between particle size and the swelling behavior of casein nanoparticles.

### 3.3. In Vitro Cytotoxiciny

#### 3.3.1. MTT Test for Cell Viability

The results of the MTT test were used to assess cell viability and determine the half-maximal inhibitory concentration (IC50) value (Figure 9A). On the 3rd hour after treatment, all of the cells maintained their viability close to the control cells (*p* < 0.05). After 24 h, the cell viability decreased and allowed for the determination of IC50—5 µg/mL daunorubicin. At 72 h, the results of the MTT test demonstrated a dramatic decrease of cell viability to 25%. 

To determine the effect of casein-based nanoparticles on the Reh cell line, DRB-loaded nanoparticles from batches Cas4-DRB-5, Cas3-DRB-5, Cas2-DRB-5, and Cas1-DRB-5 were used at concentrations corresponding to IC50 (Figure 9B). At 3 h, the viability was close to that of the control cells, but at 24 and 72 h it decreased to 30%. Blank casein nanoparticles that were used as a negative control had no statistically significant effect on cell viability. Since casein is known to have a cytoprotective effect [49], this may explain the higher viability of cells treated with casein nanoparticles compared to the DRB-free casein nanoparticles. According to the results obtained, the batch that had the most significant effect on cell vitality was Cas1-DRB-5.

#### 3.3.2. Seahorse Assay

The mitochondrial metabolism of the Reh cell line was analyzed with a Seahorse XFp instrument (Agilent Technologies, Santa Clara, CA, USA) and the oxygen consumption rate (OCR) was measured. To investigate the potential mechanism of effect of the casein nanoparticles on mitochondrial activity, a Mito stress test was performed according to the manufacturer’s protocol. The panel shows that the Reh cells exposed to nanoparticles of batch Cas1-DRB-5 demonstrated reduced levels of oxygen consumption, indicative of mitochondrial dysfunction. A decrease in maximal respiration, ATP production, and spare respiratory capacity was found. Statistical comparisons were performed based on control cells and a *p*-value < 0.05 was considered a statistically significant difference. Casein-treated cells were used as a control. The Reh cells were found to have a typical OCR mitochondrial profile, but it was reduced after nanoparticle treatment compared to the control cells as early as the 3rd hour. After one day, the results were dramatically lower (Figure 10A). 

Treatment with batch Cas1-DRB-5 resulted in a statistically significant decrease in basal respiration (from 25 to 12.3 pmol/min on the 3rd and from 23.6 to 4.2 pmol/min on the 24th hour), maximal respiration (from 30.1 to 25.9 pmol/min on the 3rd and from 24.8 to 3.2 pmol/min on the 24th hour), and ATP production (from 22.2 pmol/min to 17.1 pmol/min on the 3rd and from 20.1 to 3.2 pmol/min after 24 h). The decrease in spare respiratory capacity from 15.2 to 13.6 pmol/min was not significant after 3 h of treatment but after 24 h, decreased levels from 8.2 pmol/min to 5 pmol/min were found (Figure 10B).

In the Reh cells, DRB-loaded casein nanoparticles were found to alter the parameters of mitochondrial respiration and ATP production; furthermore, all of the observed effects were significantly pronounced. Daunorubicin is one of the most effective anticancer drugs that induces cytotoxicity by activating various mechanisms, including inhibition of DNA/RNA synthesis, heterochromatinization, oxidative stress, etc. [50]. Topoisomerase II, an enzyme that is present in both the nucleus and mitochondria of human cells, could also be inhibited. It can be assumed that the formulated casein-based nanoparticles induce an effect at the mitochondrial level. Casein, on the other hand, as a natural drug carrier, shows high efficiency for drug delivery because of its lack of toxicity and ability to improve drug bioavailability and efficacy [51]. Due to their energy-independent penetration through the plasma membrane, casein-based nanoparticles possess the ability to easily cross membranes and enhance permeability in ex vivo experiments [52]. Combining the cytotoxic potential of DRB with the advantages of casein as a carrier in the formulated DRB-loaded casein-based nanoparticles not only provides prolonged drug release but also increases permeability across the cell membranes and induces mitochondrial dysfunction, leading to a cytotoxic effect on the Reh cell line. Our results can be explained by the activation of different cytotoxic mechanisms such as increasing the generation of reactive oxygen species (ROS) and/or disrupting the integrity of the mitochondrial membrane. This leads to disruption of mitochondrial potential, inhibition of cell viability and proliferation, and ultimately cell death. 

#### 3.3.3. In Vitro Fluorescence Microscopy

In vitro fluorescence microscopy of Reh cells treated with daunorubicin-free and daunorubicin-loaded casein nanoparticles demonstrated different patterns of drug-associated fluorescence (Figure 11). During the first 3 h, strong fluorescence was observed throughout the cytoplasm and in the nucleus. Localized areas of more intense fluorescence in the cytoplasm were likely due to drug accumulation in cell organelles such as endosomes or mitochondria. The Reh cells treated with DRB-loaded casein nanoparticles demonstrated punctate, less intense fluorescence within the cytoplasm and the nucleus compared to the drug-free casein nanoparticles which correlates to the results obtained in the in vitro drug release study. 

Between the 3rd and 24th hour of treatment, increased fluorescence was observed and loss of the punctate appearance in the cytoplasm and the nucleus. The nucleus seemed to be more fluorescent than the cytoplasm, and the distribution of fluorescence throughout each cell resembled that of the DRB-free casein nanoparticles. Compared to the drug-free casein nanoparticles, the DRB-loaded nanoparticles demonstrated increased fluorescence in tumor cells after 24 h of treatment. Similar results were reported by Forssen et al. [53].

## 4. Conclusions

Within the goal of this study, daunorubicin-loaded casein nanoparticles were developed using the coacervation method followed by nano spray-drying. The particle sizes varied between 127 and 167 nm. The entrapment efficiency was highly dependent on the polymer-to-drug ratios and increased with decreasing casein concentration. The drug release was pH dependent, which confirmed our hypothesis that casein particles are suitable to be used as carriers for prolonged DRB delivery. The nanoparticles were found to alter the parameters of mitochondrial respiration and ATP production, and pronounced cytotoxicity was observed in the Reh lymphoblastic cell line.

## Figures and Tables

**Figure 1 pharmaceutics-15-00471-f001:**
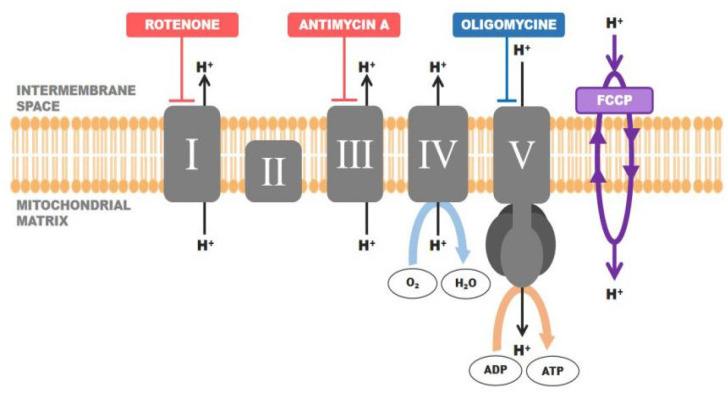
The effect of inhibitors on the mitochondrial respiratory chain in the Mito stress test. Created with Adobe Illustrator.

**Figure 2 pharmaceutics-15-00471-f002:**
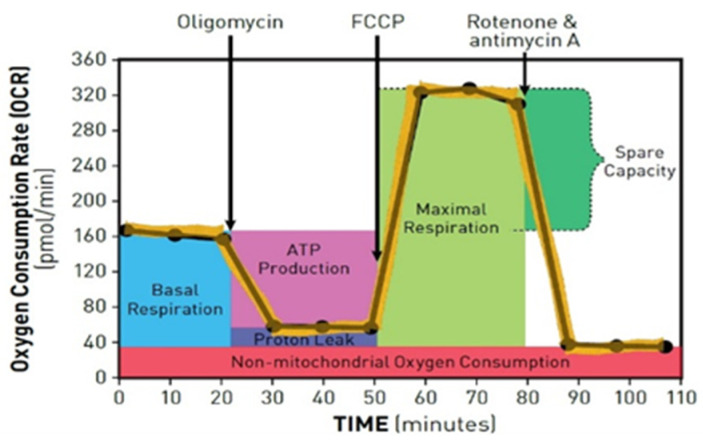
The main parameters of mitochondrial function are presented on the Mito stress test profile [35].

**Figure 3 pharmaceutics-15-00471-f003:**
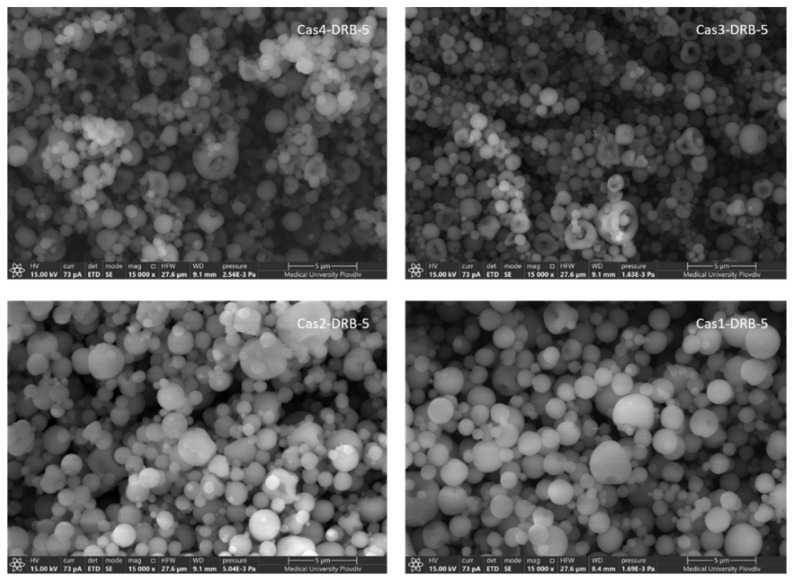
SEM micrographs of DRB-loaded casein nanoparticles of various batches at ×15,000 magnification.

**Figure 4 pharmaceutics-15-00471-f004:**
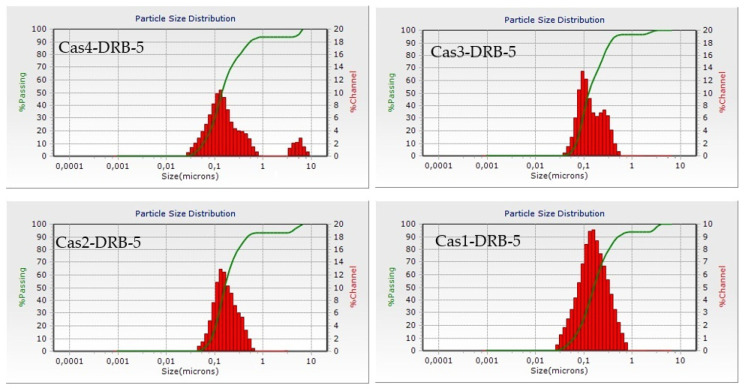
Dynamic light scattering histograms of DRB-loaded casein nanoparticles of different batches.

**Figure 5 pharmaceutics-15-00471-f005:**
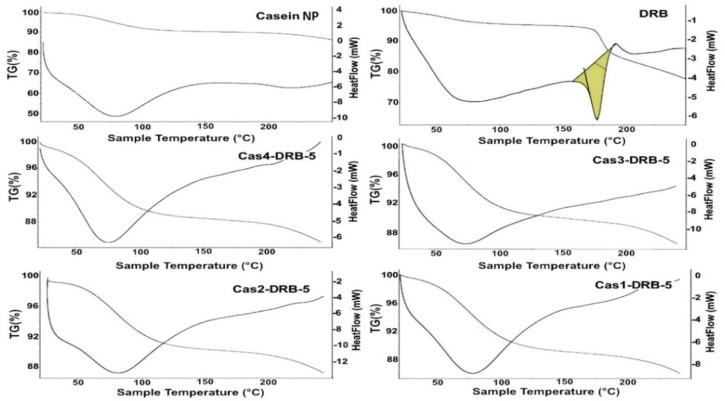
TG/DTA thermograms of blank casein nanoparticles and DRB−free and DRB−loaded casein nanoparticles of various batches.

**Figure 6 pharmaceutics-15-00471-f006:**
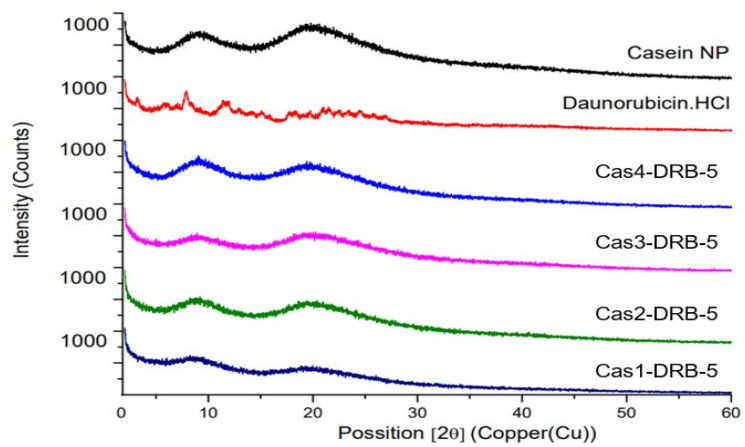
X-ray diffraction patterns of blank casein nanoparticles and DRB-free and DRB-loaded casein nanoparticles of different batches.

**Figure 7 pharmaceutics-15-00471-f007:**
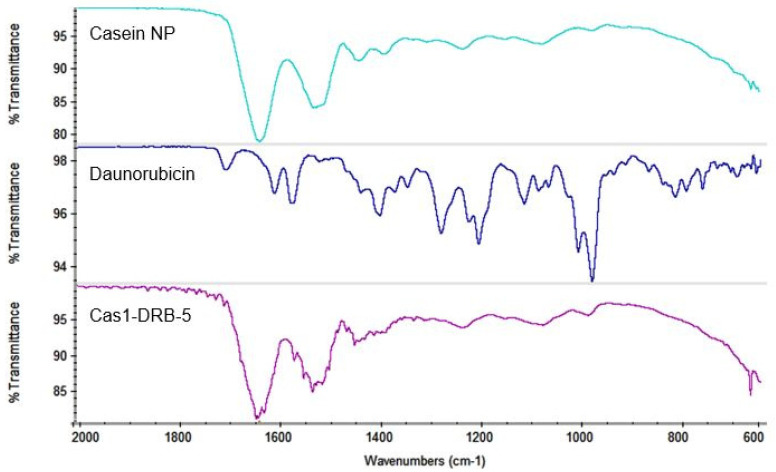
ATR-FTIR spectra of blank casein nanoparticles and DRB-free and DRB-loaded casein nanoparticles of batch Cas1-DRB-5.

**Figure 8 pharmaceutics-15-00471-f008:**
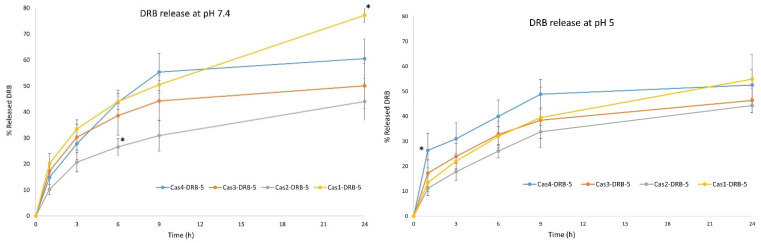
DRB release profiles from batches Cas4-DRB-5, Cas3-DRB-5, Cas2-DRB-5, and Cas1-DRB-5 at pH 7.4 and pH 5 (n = 6, *—statistically significant difference between models at the same time points, *p* < 0.01).

**Figure 9 pharmaceutics-15-00471-f009:**
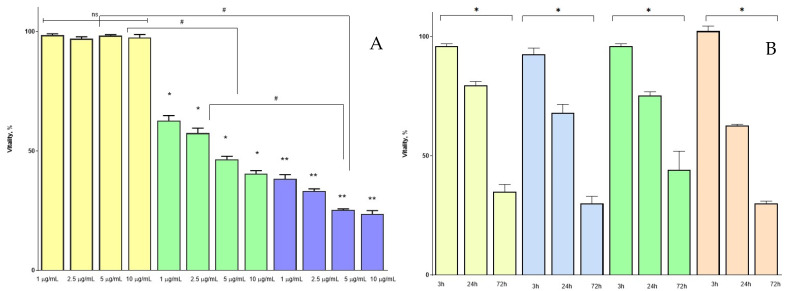
Cytotoxicity assay based on the viability of Reh cells treated with: (**A**)—increasing concentrations of DRB after 3 h (yellow bars), 24 h (green bars), and 72 h (blue bars) (n = 12, ns—no significant difference, *p* > 0.05; #—statistically significant difference between the groups, *p* < 0.05; *—statistically significant difference within the group treated for 24 h, *p* < 0.05); **—statistically significant difference within the group treated for 72 h, *p* < 0.05); (**B**)—casein nanoparticles of models Cas4-DRthe B-5 (yellow bars), Cas3-DRB-5 (blue bars), Cas2-DRB-5 (green bars), and Cas1-DRB-5 (orange bars) containing DRB equivalent of IC50 (5 µg/mL) 3 h, 24 h, and 72 h after treatment (n = 12, *—statistically significant difference within groups, *p* < 0.05).

**Figure 10 pharmaceutics-15-00471-f010:**
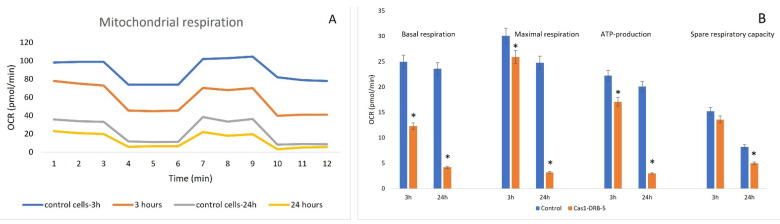
Effect of daunorubicin-loaded casein nanoparticles on cellular energetics, determined by Seahorse assay: (**A**)—representative diagram of the mitochondrial profile of the controls and the Cas1-DRB-5-treated Reh cells on the 3rd and on the 24th hour; (**B**)—bioenergetic parameters extracted from the OCR plot. Statistical analysis of the data represents the data as a mean value ± SEM; n = 3, *—*p* ≤ 0.05.

**Figure 11 pharmaceutics-15-00471-f011:**
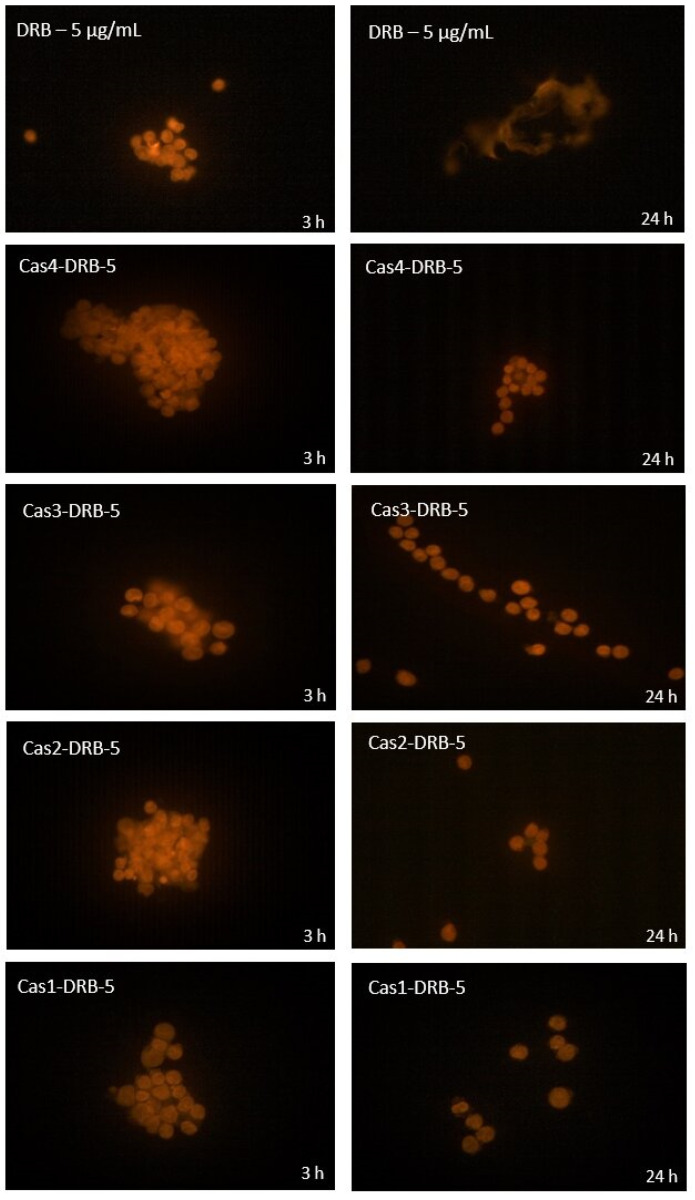
Fluorescence microscopy of Reh cells treated in vitro for 3 h and 24 h with daunorubicin-free (5 µg/mL) and daunorubicin-loaded casein nanoparticles of batches Cas4-DRB-5, Cas3-DRB-5, Cas2-DRB-5, and Cas1-DRB-5 (400× magnification).

**Table 1 pharmaceutics-15-00471-t001:** Composition of DRB-loaded casein nanoparticles of different batches.

Sample Code	Sodium Caseinate (%)	Daunorubicin HCL (mg)	Polymer:Drug Ratio
Cas4-DRB-5	2.0	5.0	100:1
Cas3-DRB-5	1.5	5.0	75:1
Cas2-DRB-5	1.0	5.0	50:1
Cas1-DRB-5	0.5	5.0	25:1

**Table 2 pharmaceutics-15-00471-t002:** Characteristics of DRB-loaded casein nanoparticles (PDI = polydispersity index, ζ = zeta potential, DL = drug loading, and EE = entrapment efficiency); n = 3.

Sample Code	Particle Size ± SD (nm)	PDI ± SD	ζ ± SD (mV)	DL ± SD (%)	EE ± SD (%)	Yield ± SD (%)
Cas4-DRB-5	167 ± 38	29.10 ± 3.88	−33.21 ± 3.98	2.14 ± 0.07	42.80 ± 0.32	37.67 ± 1.32
Cas3-DRB-5	162 ± 37	25.06 ± 4.02	−25.53 ± 5.64	2.38 ± 0.06	47.60 ± 0.09	46.91 ± 1.27
Cas2-DRB-5	142 ± 13	27.4 ± 3.43	−21.52 ± 3.41	2.97 ± 0.07	59.04 ± 0.22	79.63 ± 1.45
Cas1-DRB-5	127 ± 42	7.65 ± 2.02	−18.63 ± 3.37	3.09 ± 0.11	61.80 ± 0.16	81.12 ± 1.58

## Data Availability

The data presented in this study are available on request from the corresponding author.

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
