# Peer review of "Casein-Based Nanoparticles: A Potential Tool for the Delivery of Daunorubicin in Acute Lymphocytic Leukemia"

_pharmaceutics, 2023, doi:10.3390/pharmaceutics15020471_

Round 1

Reviewer 1 Report

This research focuses on preparing casein nanoparticles by spray dryer for drug delivery applications. The authors have synthesized casein nanoparticles using SEM, XRD, FT-IR, DLS, and TG/DTA The article has many grammatical and sentence errors, and the language organization needs to be improved. For these reasons, I conclude that the paper is suitable for publication with a major revision.

1. Authors need to discuss the importance of targeted drug release and controlled drug release in cancer treatment. 

2. Authors can add a note on the advantages of proteins over metal/ metal oxide nanoparticles. refer and cite ,https://doi.org/10.3390/pharmaceutics12070604, https://doi.org/10.1016/j.lwt.2021.112953  

3. Also mention various other proteins used in drug delivery applications. who casein is better than other proteins in biomedical applications., such as cheap, its reducing ability for nanoparticle preparation, etc.  Refer https://doi.org/10.1016/j.jconrel.2011.02.010, https://doi.org/10.3390/nu14193888, https://doi.org/10.1016/j.ijpharm.2019.118652

4. In MTT assay, authors may free drug as a control and bare casein nanoparticles as a negative control for comparison of how loading improved the efficacy of the drug 

5. IC50 value for all the formulations needs to calculate in a table 

6. Typographic errors need to be corrected. The language and grammar used throughout the manuscript need to be improved

Author Response

Responses to Reviewer 1

We appreciate the reviewer valuable comments and recommendations. The responses to Reviewer remarks are provided below. We hope that they are going to satisfy the Reviewer:

Point 1

Authors need to discuss the importance of targeted drug release and controlled drug release in cancer treatment. 

Response 1

Although we have outlined the concept of targeted delivery and its importance in cancer treatment, we have extended the discussion by adding a new paragraph (line 40).

Point 2

Authors can add a note on the advantages of proteins over metal/ metal oxide nanoparticles. refer and cite ,https://doi.org/10.3390/pharmaceutics12070604, https://doi.org/10.1016/j.lwt.2021.112953  

Response 2

We appreciate the reviewer’s suggestion. However, with this work we did not aim to compare protein and metal oxide NPs. In our research group, we are investigating both types of delivery systems and we believe that each has its own advantages and disadvantages. However, we used the proposed reference to reinforce the importance of proteins as carriers.

Point 3

Also mention various other proteins used in drug delivery applications. who casein is better than other proteins in biomedical applications., such as cheap, its reducing ability for nanoparticle preparation, etc.  Refer https://doi.org/10.1016/j.jconrel.2011.02.010, https://doi.org/10.3390/nu14193888, https://doi.org/10.1016/j.ijpharm.2019.118652

Response 3

A new paragraph was added (line 63) and the proposed references were cited.

Point 4

In MTT assay, authors may free drug as a control and bare casein nanoparticles as a negative control for comparison of how loading improved the efficacy of the drug 

Response 4

The results of the MTT test were used to determine Reh cell viability after treatment with free Daunorubicin (Figure 9A), and with casein-based nanoparticles (Figure 9B).  Based on the results shown in  Figure 9A we determined the Daunorubicin inhibitory concentration value (IC50 = 5 ug/mL) and after that, we used amounts of the batches corresponding to this IC50 concentration (Figure 9B). As a negative control, we used blank casein nanoparticles in concentrations that correspond to the maximal per cent of casein in our models - 2%. Due to the fact that casein has no statistically significant effect on cell viability at all time points and to make the figure clearer, we didn't show the results. On the Figure 9 the results are presented as relative per cent value.

Point 5

IC50 value for all the formulations needs to calculate in a table 

Response 5

With respect to the reviewer’s opinion, we have determined IC50 for Daunorubicin only as an API. Then, we have treated the cells with a defined amount of NPs, containing the required drug amount so that we achieve the desired concentration.

Point 6

Typographic errors need to be corrected. The language and grammar used throughout the manuscript need to be improved.

Response 6

We have revised the manuscript for typos. We hope that the language and grammar were improved.

Reviewer 2 Report

I read this article with a great interest.

the title is clearly missleading : ALL "cell line" should be added the main aim of the article being on the formation of the nanoparticules with casein.

it is clearly a first step towards less toxicity by daunorubicin which is a real problem.

the results are well presented but do not treat the toxicity problem which is raised in the introduction.

In the discussion more data are required concerning the choice of caseine and the authors should insist more on the eventual role on toxicity of their new treatment (casein-based nanoparticules) 

the conclusion should be modified as the results are only in vitro data

Author Response

Responses to Reviewer 2

We appreciate the reviewer valuable comments and recommendations. The responses to Reviewer remarks are provided below. We hope that they are going to satisfy the Reviewer:

Point 1

the title is clearly missleading : ALL "cell line" should be added the main aim of the article being on the formation of the nanoparticules with casein.

Response 1

We have modified slightly the title emphasizing on the potential use of the formulated nanoparticles in ALL therapy.

Point 2

it is clearly a first step towards less toxicity by daunorubicin which is a real problem.

the results are well presented but do not treat the toxicity problem which is raised in the introduction.

Response 2

We fully agree with the reviewer that cardiac toxicity is an important concern for anthracycline antibiotic chemotherapeutics. However, in our study, we aimed to develop an optimised daunorubicin-loaded delivery system focusing on efficacy. Our main goal was to investigate whether incorporation of DRB into casein nanoparticles would benefit their cytotoxic effect against tumor cells, not on healthy cells. In our upcoming studies, this could be our next target. We thank to the reviewer for this suggestion.

Point 3

In the discussion more data are required concerning the choice of caseine and the authors should insist more on the eventual role on toxicity of their new treatment (casein-based nanoparticules) 

Response 3

Data regarding the advantages of casein compared to other biopolymers was provided in the Introduction. Furthermore, we

Point 4

the conclusion should be modified as the results are only in vitro data

Response 4

We apologise if we misunderstood the reviewer’s remark. The conclusion summarizes the main findings of the study. We have not claimed anything other than in vitro studies.

Round 2

Reviewer 1 Report

All the queries have been rightfully addressed. Recommended for publication  

Reviewer 2 Report

the authors did answer my questions. Thanks